# Application of Polymer Curing Agent in Ecological Protection Engineering of Weak Rock Slopes

**Ding Yao [1,3,*], Guoping Qian [1,2], Jiawu Liu [1,4] and Jialiang Yao [1,2]**

[1]  School of Traffic and Transportation Engineering, Changsha University of Science & Technology, Changsha 410114, China; guopingqian@sina.com (G.Q.); xiao0833@126.com (J.L.); yao26402@126.com (J.Y.)
[2]  Key Laboratory of Special Environment Road Engineering of Hunan Province, Changsha University of Science & Technology, Changsha 410114, China
[3]  Hunan Dingniu Engineering Consulting Co., Ltd., Changsha 410015, China
[4]  Hunan Huida Planning, Survey and Design Research Co., Ltd., Changsha 410004, China
*  Correspondence: yao9206@stu.csust.edu.cn; Tel.: +86-151-1101-0089

**Abstract:** Under the action of water, weak rock slopes easily expand and the strength and stiffness decrease, which results in slope instability. The styrene–acrylic emulsion cement-matrix composite, a new type of polymer curing agent, was developed for the curing and treatment of weak rock slopes. The strength-reduction factor method and ANSYS finite element software were used to calculate and analyze the stability of slopes before and after protection. The stability safety factor of weak rock after protection increased by 30% from 2.0 to 2.6. In order to evaluate the performance of the polymer curing agent, the mixture test was carried out in the laboratory. It was found that the waterproofness, hydrophobicity, and microstructure of weak rock slopes with the polymer curing agent can be significantly improved. Finally, the polymer curing agent was adopted and the external-soil spray-seeding technique was used in physical engineering. From test results, it was indicated that the polymer curing agent for weak slopes is beneficial in improving the water-damage resistance of a slope surface and prevent or reduce the softening of weak rock so that plants can grow for a long time. The treatment for weak rock slopes was successfully combined with plant protection, achieving the dual effect of weak rock slope protection and ecological protection.

**Keywords:** polymer curing agent; weak rock; slope curing; ecological protection; stability; engineering application

## 1. Introduction

Water is one of main factors in the instability of weak rock slope, which has a significant influence on the balance of the internal structure of rock mass [1–3]. In essence, the softening and instability collapse of weak slopes are mainly attributed to the interaction of internal structures of water–weak rock mass. Craw et al. believed that the interaction between water and weak rock was mainly the process of dissolution–reprecipitation and physical and chemical reactions, such as ion exchange. In this process, the increase in volume and the decrease in the strength of rock mass could result in slope instability [4–6]. Some scholars have studied the saturated and unsaturated seepages of rock mass fissures through numerical simulation. It has been shown that the seepage of rock mass changed greatly and expanded continuously, which had great influence on the stability of the slope [7–10]. Water can destroy the tiny minerals and pore structures on the surface of the rock mass, thus affecting the stable microstructures of the rock mass. These mineral and microstructural changes can cause shallow slope damage. When water exists in the slope for a long time, it can reduce the ultimate compressive pressure of the rock mass, which eventually results in deep creep deformation [11–13].

Therefore, as long as the contact between the water and the weak slope surface is prevented or reduced, the softening of the weak slope can effectively be slowed down.

At present, traditional masonry protections, such as grouted rubble masonry, dry rubble masonry, and spray concrete with steel bar nets, have been generally adopted in weak rock slope protection worldwide. Although weak rock slope could effectively be prevented from collapsing, the strength and efficiency of protection has been excessively pursued resulting in the destruction of the harmony of natural ecology so that green trees and clean rivers have been replaced by the hard and rigid "cement channel" that was constructed in slope protection engineering [14–16]. The construction of landscape effects and the normal growth of plants at that time were mainly focused on the nowadays commonly used methods of ecological protection of weak rock slope. However, the denudation and softening of weak rock slope and the overall structural instability could not be prevented [17–19]. The curing materials for these slopes have developed from the original single cement, fly ash, or lime into the new curing materials, such as the slope-curing agent. The slope-curing agent refers to the composite material which can improve the stability of engineering slope. It can also overcome and improve the shortcomings of single materials, such as cement, fly ash, and lime. Most studies focus on the solidification and modification of soil slopes [20–22]. However, the polymer curing agent may have different effects on the ecological protection of weak rock slopes after it is used for slope curing.

The new polymer curing agent was mainly proposed in terms of hydration disintegration, water softening, strength reduction, and other characteristics of weak rock. It can prevent, weaken, or slow down the softening, disintegration, and collapse of weak rock slopes, thus improving the long-term stability. The external-soil spray-seeding technology with microbial polymer water-absorbing material was carried out in the green ecological protection of slopes, which played a role in improving the stability of weak slopes and thus created a safe, green, and environmentally friendly road operation environment.

## 2. Stability Analysis of Weak Slope under the Protection of Polymer Curing Agent

The instability of slopes caused by rainwater landslides and collapses in weak slopes has drawn widespread attention. Due to the complexity of the problem, it is important to understand how to mitigate the nature of the softening problem of soft rock slopes. From the perspective of preventing or reducing the contact opportunity between water and soft rock slope surfaces, the performance requirements of a polymer curing agent should be proposed for the non-curing agent protection of soft rock slopes and the finite element analysis under the protection conditions of a new polymer curing agent.

### 2.1. Mohr–Coulomb Failure Criteria

In the non-rain period, weak rock is usually unsaturated. In addition, suction exists in the weak rock, which maintains the stability of slope. However, as the rain permeates, the water content of soil mass increases. The suction gradually decreases and cohesion is reduced, which leads to the decrease in the shear strength of the rock mass [23–26]. When the shear stress of the rock mass exceeds the shear strength of soil mass itself, the soil will slide along one of its sliding surfaces, thus causing the soil to lose overall stability. This is according to the Mohr–Coulomb strength theoretical formula, $|\tau_f| = c' + \sigma' tan\varphi'$, where the cohesion, $c'$, effective normal stress, $\sigma'$, and the internal friction angle, $\varphi'$, are main influencing factors.

### 2.2. Finite Element Analysis of Polymer Curing Agent Protection

The Hunan experimental slope was taken as an example and the finite element software was used to analyze the stress, strain, and stability of weak rock slopes. The slopes without protection and with the new polymer curing agent were tested. The strength reduction method was used for the slope stability calculation. Based on the associated flow rule, the dilatancy angle, $\Psi = \varphi$, was selected in the finite element analysis. Firstly, the initial reduction factor, $F$, was selected and then the strength

coefficient of the materials of the slope soil mass was reduced. The cohesive force, $C_n$, and the internal friction angle, $\varphi_n$, were calculated using Formula (1) and Formula (2) respectively.

$$C_n = \frac{C}{F} \tag{1}$$

$$tan\varphi_n = \frac{tan\varphi}{F} \tag{2}$$

The reduction factor, $F$, was input into the slope model. If it converged, it indicated stability of the slope. It continued to increase until the program did not converge, which shows the stability or safety factor [27–29].

### 2.2.1. Engineering Overview of Test Slope

The location of the slope was at the left side of the K1141 + 540 – K1141 + 580 section of the Lianyuan–Zhuzhou Highway. The highest point of slope was 30 m high. It was excavated in three stages, with a slope ratio of 1:0.75. Each stage slope was 10 m high and the first and second platforms were 2 m wide. It was a typical weak rock slope, mainly composed of intense weathering granite. The section of weak rock slope engineering is shown in Figure 1.

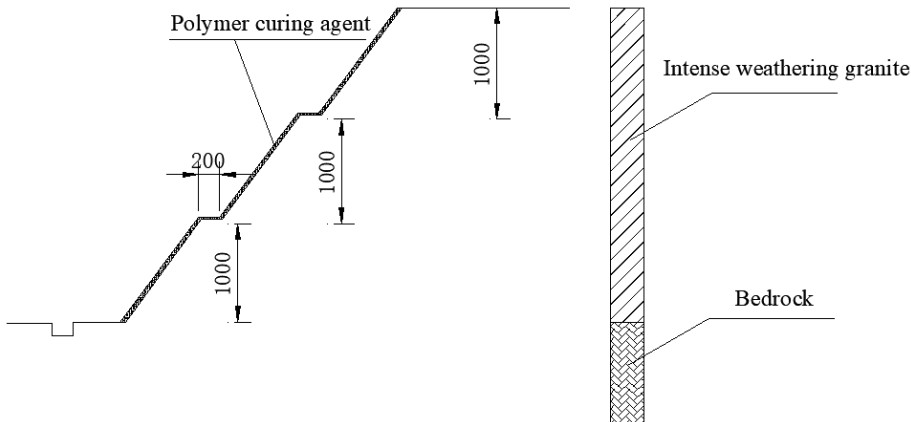

**Figure 1.** Slope engineering section.

### 2.2.2. Calculation Parameters and Model of Weak Rock Slope

(1)    Weak rock slope parameters

In the slope simulation analysis, the finite element software was used and the model of geomaterials was generally defined as a DP model. Therefore, the weak rock materials were defined as ideal elastoplastic materials and the bedrock materials were defined as elastic materials. The yield condition for simulating elastoplastic property was based on the Mohr–Coulomb criterion and the plane model with plane strain analysis was taken as the slope model. During the calculation, the elastic deformation of the medium itself was taken into account. Therefore, it could truly reflect the stress and strain of slope under the stressed state and the unit in the lower part of the plastic zone could produce certain vertical deformation and horizontal deformation, basically eliminating the plastic zone in the lower part of slope due to the boundary effect. The deformation of the slope and the development of the plastic zone were well simulated [30–32]. Material parameters used in the slope stability analysis and geotechnical parameters under different reduction factors are shown in Table 1 below.

**Table 1.** Material parameters of slope model.

| Material Type | $E$ (GPa) | $v$ | $\gamma$ (kN/m³) | $C$ (MPa) | $\Phi$ (°) |
|---|---|---|---|---|---|
| Intense weathering granite | 0.57 | 0.23 | 24.6 | 0.140 | 34.3 |
| Polymer curing agent + intense weathering granite | 1.00 | 0.25 | 23.5 | 0.200 | 45.8 |
| Bedrock | 20.00 | 0.32 | 26.0 | 1.500 | 50 |

(2)   Establishment of weak slope model

The slope stability analysis model was a three-level slope with a width of 46.5 m and a height of 38 m. The geometry of the slope is shown in Figure 2a. The slope model was divided into 1139 units and 3111 nodes. The 8-node PLANE183 unit was used. Each unit has plasticity, creep, expansion, stress stiffening, large deformation, and large strain. Each node has two degrees of freedom, UX and UY. The boundary constraints of the slope model were set as follows: The lateral restraint was used on both sides of the slope, that is, horizontal displacement was not allowed on both sides. In order to limit horizontal displacement and vertical displacement at the bottom of slope, full displacement constraints were set here. The mesh division and the displacement constraint diagram of the slope are shown in Figure 2b.

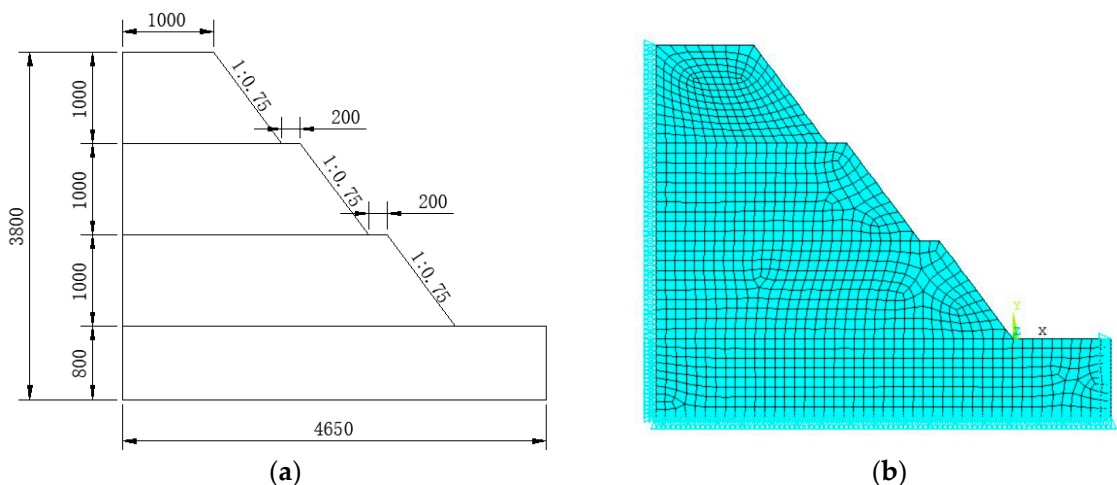

|              |              |
|:------------:|:------------:|
| (**a**)      | (**b**)      |

**Figure 2.** Slope model: (**a**) Slope geometry; (**b**) mesh division and displacement constraint of the slope.

2.2.3. Load Selection and Stability Analysis Design

It was possible to realistically simulate the change of the slope under the action of self-heavy stress by applying a vertical gravitational acceleration (G = 9.8 m/s²) to the slope model [33,34]. In this model, the strength reduction finite element method was used to analyze the stability of slope. The shear strength parameters of the slope rock and the soil mass were gradually reduced until the slope reached the ultimate failure state. According to the finite element calculation results, the slip failure surface was automatically obtained by the program and the strength storage safety factor of slope was calculated. The stability of the rock and the soil mass before and after the slope protection was analyzed and solved by changing the strength reduction factor. The horizontal displacement, vertical displacement, and strength reduction safety factor of rock mass before and after protection were obtained. The material values of the rock and soil mass are shown in Table 2 below.

**Table 2.** Parameter values of geotechnical materials under different reduction coefficients.

| Reduction Factor $F$ | $E$ (GPa) | $v$ | $\gamma$ (kN/m³) | $C_n$ (MPa) | $\varphi_n$ (°) | Remarks |
|---|---|---|---|---|---|---|
| 1.2 | 0.57 | 0.23 | 24.6 | 0.117 | 28.6 | |
| 1.4 | 0.57 | 0.23 | 24.6 | 0.100 | 24.5 | |
| 1.6 | 0.57 | 0.23 | 24.6 | 0.088 | 21.4 | Does not set |
| 1.8 | 0.57 | 0.23 | 24.6 | 0.078 | 19.1 | curing agent |
| 2.0 | 0.57 | 0.23 | 24.6 | 0.070 | 17.2 | layer |
| 2.2 | 0.57 | 0.23 | 24.6 | 0.064 | 15.6 | |
| 1.2 | 1.00 | 0.25 | 23.5 | 0.167 | 38.2 | |
| 1.4 | 1.00 | 0.25 | 23.5 | 0.143 | 32.7 | |
| 1.6 | 1.00 | 0.25 | 23.5 | 0.125 | 28.6 | |
| 1.8 | 1.00 | 0.25 | 23.5 | 0.111 | 25.4 | Set curing |
| 2.0 | 1.00 | 0.25 | 23.5 | 0.100 | 22.9 | agent layer |
| 2.2 | 1.00 | 0.25 | 23.5 | 0.091 | 20.8 | |
| 2.4 | 1.00 | 0.25 | 23.5 | 0.083 | 19.1 | |
| 2.6 | 1.00 | 0.25 | 23.5 | 0.077 | 17.6 | |
| 2.8 | 1.00 | 0.25 | 23.5 | 0.071 | 16.4 | |

### 2.2.4. Deformation and Stability Analysis of Slope

The strength reduction finite element method was used to analyze the rock and soil mass with protection and without protection. The slope under natural conditions was taken as the initial slope and its deformation law under the gravity stress was analyzed [35–37].

(1)    Deformation law of weak slope without the protection of curing agent under natural conditions

The unprotected slope in the natural state was analyzed and the horizontal displacement (X-direction displacement) and the vertical displacement (Y-direction displacement) were obtained. The displacement nephograms in the X and Y directions are shown in Figure 3.

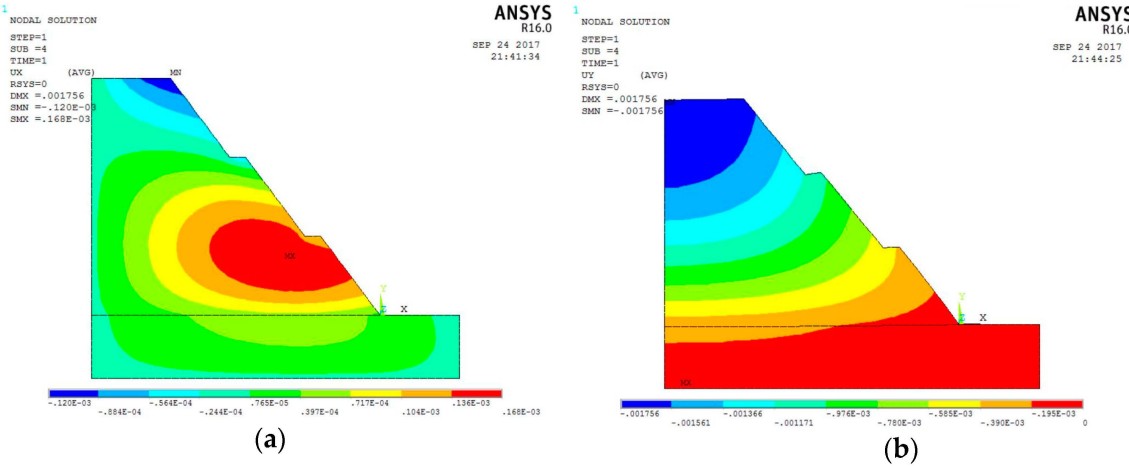

**Figure 3.** Displacement nephograms: (**a**) X direction; (**b**) Y direction.

From the calculation results of ANSYS, it can be seen that the slope constitutive model is in a state of convergence in the initial state. Therefore, the slope is in a safe state. From the analysis of the calculated X- and Y-direction displacement nephograms, it can be found that the horizontal displacement of the first-stage slope wall gradually increased with the increase of slope height. The horizontal displacements at the foot and the top of slope are 7.6 mm and 17.6 mm, respectively. The horizontal displacements of the horizontal wall of the second- and third-stage slopes gradually decreased with the increase of the slope height. They were 8.6 mm and 6.3 mm, respectively, at the foot of slope and both 4.3 mm at the top of slope. They were different in the inner side and outer side of the

slope platform. The horizontal displacement of the outer side was larger than that of the inner side. The settlement rate of the surface settlement displacement in the vertical direction was related to the vertical height. The higher the height was, the slower the settlement rate was. The settlement value was inversely proportional to the height. When the height increased, the settlement value gradually decreased. The maximum settlement value at the top of slope was 3.1 mm.

(2)    Stability analysis of weak slope before and after protection by reduction factor method

The cohesion and internal friction angles of the slope rock and the soil mass were changed by changing the reduction factor. Different deformations of the rock and the soil mass with protection and without protection were simulated. According to the ANSYS software, the results of calculation and analysis are shown in Figures 4 and 5.

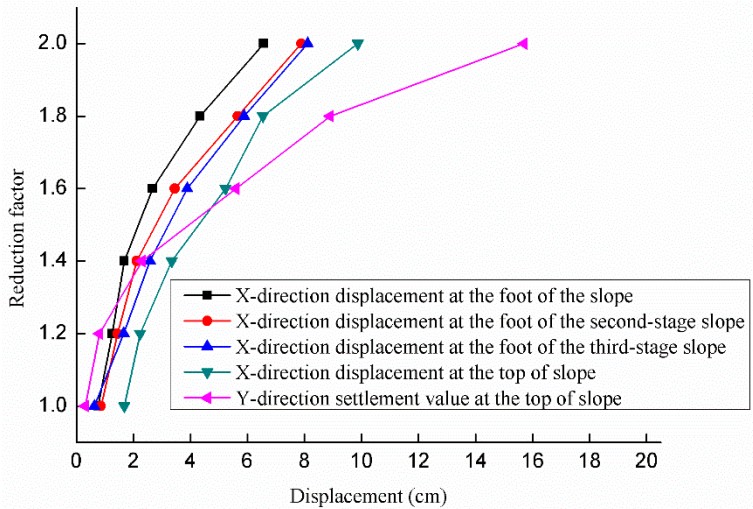

**Figure 4.** X-direction and Y-direction displacements of the unprotected slope with a changed reduction factor.

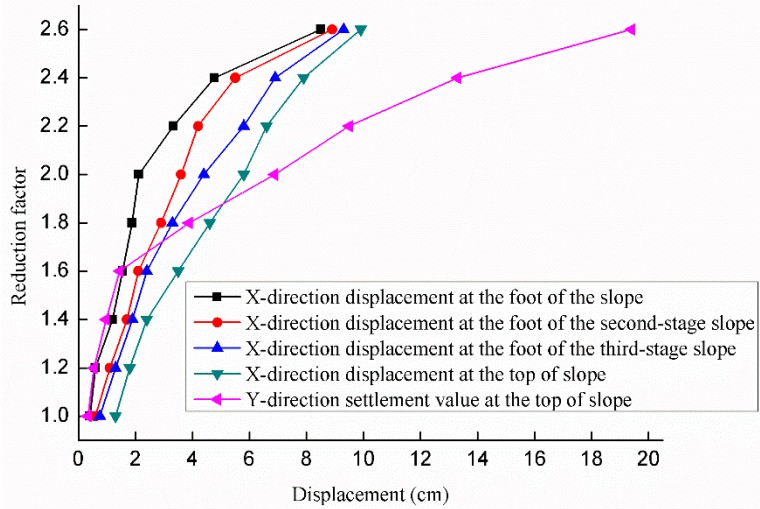

**Figure 5.** X-direction and Y-direction displacements of the protective slope with a changed reduction factor.

According to the protection methods used in the physical engineering and the material parameters obtained from the indoor test, the finite element analyses of the weak rock slope with the protection of the new polymer curing agent and without it were carried out by changing the strength reduction factor respectively. Analysis results are as follows:

(1) The weak slope without the protection of curing agent was no longer converged when the reduction factor was greater than 2.0 and the displacement in the vertical direction increased sharply, thus indicating that the failure appeared in the slope at this time. Therefore, the safety factors of the slope without protection and with the protection of the curing agent were 2.0 and 2.6, respectively.

(2) The change rate of displacement in the X and Y directions of the weak slope with the curing agent was smaller than that of the slope without curing agent. Therefore, the protection design of the polymer curing agent should focus on improving the softening and hydration resistance of the soft rock.

(3) After the strength reduction parameters were changed, the finite element convergence was comparatively analyzed. It was found that the polymer curing agent protection was a more practical method to improve the slope stability.

(4) From the above mentioned, it was found that the cohesive force and internal friction angle of the weak rock decreased and its strength reduction factor increased. Not only was the shear performance reduced, but also the possibility of a landslide was greatly increased. Therefore, the polymer curing agent protection was used for the weak slope. It could prevent or slow down the softening of the weak rock slope, improve the slope stability, and prolong its service life, which has important practical significance.

## 3. Materials and Tests

### 3.1. Raw Materials

The polymer curing agent is composed of styrene–acrylic emulsion with waterproofness, Portland cement, water, and a small amount of defoaming agent. It is a polymer-matrix cement composite material, which can effectively inherit the main performance of polymer emulsion and cement. As a kind of curing material for the weak rock slope, the polymer curing agent has the following unique advantages: (i) It can form a membrane structure with waterproofness, hydrophobicity, and high strength on the surface of a weak rock slope; (ii) its organic components are beneficial for plant growth.

#### 3.1.1. Polymer

The styrene–acrylic emulsion was used in this experiment, which complied with the provisions of *Emulsion for Architectural Coatings* (GBT 20623-2006). It has certain gas permeability, good adhesion, and weather resistance. It is also non-toxic, tasteless, and environmentally friendly. Test results of its technical index are shown in Table 3.

**Table 3.** Test results of the technical index of styrene–acrylic.

| NO | Test Item | Test Result | Specification |
|----|-----------|-------------|---------------|
| 1 | Appearance | Milky liquid with slight blue light | Milky liquid with slight blue light |
| 2 | Solid content/% | 53.5 | 54.0 ± 1.0 |
| 3 | pH | 7.0 | 7.0–9.0 |
| 4 | Residual stability/% | 0.02 | ≤1.0 |
| 5 | Glass transition temperature/°C | −18 | —— |

#### 3.1.2. Cement

The performance index of cement should meet the requirements of *Common Portland Cement* (GB 175-2007). If necessary, the anti-sulfate cement could be used and its strength should be greater than 32.5MPa. High-aluminum cement should not be used. A permanent slope support for the mountain was taken as the test object, so the P.O42.5 ordinary Portland cement was used.

### 3.1.3. Organic Silicone Defoamer

The technical index of the organic silicone defoamer should meet the requirements of *Organic Silicone Defoamer* (GB/T 26527-2011); its test results are shown in Table 4.

**Table 4.** Test results of the technical index of silicone defoamer.

| NO | Test Item | Test Result | Specification |
|----|-----------|-------------|---------------|
| 1 | Appearance | Yellowish | White or yellowish at room temperature |
| 2 | pH | 7 | 5.0–8.5 |
| 3 | Solid content/% | 30.1 | ≥10 |
| 4 | Defoaming time (s) | 10 | <15 |
| 5 | Stability/mL | 0.01 | ≤0.5 |

### 3.2. Curing Mechanism of Polymer Curing Agent

The polymer curing agent can prevent or reduce the softening effect of water on weak rock slopes, which is a serious problem currently. When it is sprayed onto the weak rock slope, a waterproof structural layer is formed. At the same time, the damaged surface of weak rock slope can be wrapped and bonded to re-cure into a whole. The action mechanism of the polymer curing agent and the weak rock surface is as follows [38–42]:

Waterproofness: After the evaporation of water, a polymer layer is formed by the polymer curing agent with macromolecule. It can be intertwined with the hydration product of cement to form a network structure layer, which effectively prevents water molecules from penetrating into the interior of the rock mass.

Bonding: An acid–base affinity is formed to increase the bonding when the hydroxyl (-OH) and carboxyl (-COOH) groups in the polymer are combined with a large amount of cations ($Ca^{2+}$, $Mg^{2+}$, $Al^{3+}$) on the surface of the cement and the weak rock. At the same time, the polymer curing agent with good cementation penetrates into the interior of the pores of the rock mass, which effectively increases the bonding strength of the fractured rock masses and re-cures them into a new whole.

Hydration: When the polymer emulsion is added to the cement, the polymer particles are uniformly dispersed into the cement slurry during the stirring process. When the cement is mixed with water, the hydration reaction begins. The $Ca(OH)_2$ solution rapidly reaches supersaturation and precipitates crystals. At the same time, ettringite crystal and hydrated calcium silicate gelinite are formed. Polymer particles in the emulsion are deposited on the gelinite and unhydrated cement particles.

Polymerization: The polymer curing agent sprayed on the surface of a weak rock slope can make the gravel or the fracture surface of the weak rock bond well because it can polymerize to form a polymer chain layer. It has the properties of high strength, good flexibility, and elasticity.

### 3.3. Design of Mix Proportion

Mix proportions of five kinds of curing agent were selected. The workability, hydrophobicity, bonding strength, waterproofness, and microscopic properties of the polymer curing agent were tested by physical and chemical analysis methods, such as the rotational viscosity test, the contact angle test, the adhesion test, the water-absorbing rate test, and scanning electron microscopy.

Under the same test conditions, the difference of properties was tested by adjusting the polymer–cement ratio of the polymer curing agent. Based on the test data, the optimum polymer–cement ratio was analyzed and determined, providing technical support for engineering application. The mix proportion was shown in Table 5.

**Table 5.** Mix proportion of polymer curing agent.

| NO | Styrene–Acrylic Emulsion | Solid Content of Emulsion | Cement | Polymer–Cement Ratio | Defoamer |
|---|---|---|---|---|---|
| 1-1 | 53.1 | 25 | 50 | 0.5 | 1% |
| 1-2 | 106.2 | 50 | 50 | 1.0 | 1% |
| 1-3 | 159.2 | 75 | 50 | 1.5 | 1% |
| 1-4 | 212.3 | 100 | 50 | 2.0 | 1% |
| 1-5 | 265.4 | 125 | 50 | 2.5 | 1% |

*3.4. Test Methods*

3.4.1. Workability Test

The workability of the polymer curing agent refers to the fluidity during the construction process, that is, the viscosity of the curing agent, which represents the friction of molecules inside the fluid. The fluidity embodies the workability of the curing agent, which has a great influence on the construction process of the curing agent and directly affects the protection effect and quality of engineering. The rotational viscometer verification regulation of the *National Measurement Standard* JJG1002-2005 was adopted to test the viscosity of curing agent under different polymer–cement ratios.

3.4.2. Bonding Strength Test

The bonding strength between the curing agent layer and the surface of weak rock is an important index for evaluating the performance of curing agent. After the excavation of weak rock mass, the internal and external stresses result in the deformation of rock mass and a certain degree of deformation of curing agent.

The curing agent of slope not only withstands the stress of the weak rock, but also withstands the stress of the external soil. If the curing agent layer can firmly adhere to the surface of weak rock, whose loose particles are bonded together, the internal and external stresses will be overcome, thus improving the service life of the layer. Therefore, to a large extent, the feasibility and reliability of application of curing agent slope is determined by the bonding strength of layer. In this test, the bonding strength of the coatings 7d and 28d was determined by the pull-off test, which refers to the test method of tensile bonding strength of mortar in the *Test Method of Performance on Building Mortar* JGJ/T70-2009.

3.4.3. Waterproof Test

The polymer curing agent sprayed on the weak rock slope is mainly used for preventing natural rainfall and runoff water from the surface. Its waterproof performance can be expressed by the natural water-absorbing performance, which refers to the mass ratio of the test specimen after immersion in an aqueous solution for 48 h under standard atmospheric pressure to test a specimen after drying. According to the test procedure of water-absorbing in T0205-2005 of *Test Methods of Rock for Highway Engineering* JTGE41-2005, the test procedure was carried out. During the test, the rupture condition of rock was observed and its rupture site and time were recorded in time.

3.4.4. Hydrophobicity Test

The hydrophobicity of the polymer curing agent refers to the repulsive characteristic of molecules and water molecules on the surface of curing material. Based on the surface wetting characteristic of the weak rock itself, the softening process of weak rock can be understood as the interaction between weak rock and water. The measuring instrument of DSA100 was used to measure the contact angle of droplets on the surface of weak rock and curing weak rock sprayed with the polymer curing agent. Through the comparison of contact angles before and after, the changes of hydrophobic property of weak rock surface before and after modification are effectively shown in Figure 6. The deionized or distilled water was preferably used for the measurement because highly ionized water may have

caused the test to fail. In addition, it should not have contained hydrophobic or hydrophilic substances. This experiment was carried out in dry air at 20 °C.

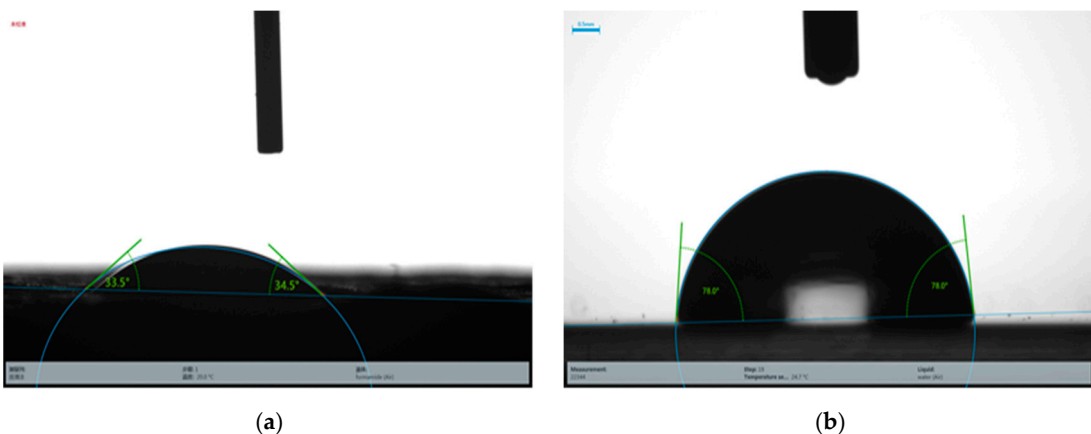

| (**a**) | (**b**) |

**Figure 6.** Contact angle of rock specimen before and after modification: (**a**) Virgin; (**b**) modification.

### 3.4.5. Microscopic Performance Test

According to test requirements, the specimen of the weak rock was prepared. The microstructure of the specimen sprayed with the polymer curing agent before and after modification was captured by scanning electron microscopy (SEM). Micrographs were obtained at 1500x magnification under an S-3000N SEM.

### 3.4.6. Test Section

Due to the excavation of subgrade, the originally buried underground rock mass was exposed to the outside so that it was directly in contact with the rainwater in the natural body, which penetrated into the inner softening rock mass through its joint fissures. Its volume slightly expanded after it encountered the water. The water-scouring resistance was weakened and the shear strength of the rock mass was reduced so that the rock mass could not withstand the external stress and slide in a slope direction. In order to obtain the engineering performance of the polymer curing agent in the ecological protection of the weak rock slope, two test slopes were established on highways of Zhejiang Province and Hunan Province, China. The Zhejiang test slope is located at the left side of the K6 + 467.3 – K6 + 597.3 section of the secondary road from Zhulanjiao to Pengxi in Wenzhou. The highest point of slope is 50 m high. It is excavated by five stages. The first-stage slope is 10 m high with a ratio of 1:0.5. The second-stage slope is 10 m high with a ratio of 1:0.75. The third-stage slope is 10 m high with a ratio of 1:1. The fourth-stage slope is 12 m high with a ratio of 1:1. The fifth-stage slope is 8 m high with a ratio of 1:1. It belongs to the mid-subtropical monsoon climate. In addition, the weather conditions are considered to be very serious. The rainfall is abundant and the annual precipitation is 2494 mm. The Hunan test slope is located at the left side of the K1141 + 540 – K1141 + 580 section of the Lianyuan–Zhuzhou Highway. The highest point of the slope is 30 m high. It is excavated in three stages with a slope ratio of 1:0.75 and a height of 10 m. It belongs to the subtropical monsoon humid climate. In addition, the weather conditions are considered to be very serious. The rainfall is abundant and the annual precipitation is 1719 mm. The Protection design was shown in Table 6.

**Table 6.** Protection design of the test slope.

| Test Section | Rock | Geological Characteristics | Treatment Location | Protection Plan |
|---|---|---|---|---|
| Zhejiang test slope | Highly weathered tuff of crystal and vitric fragment | It is light gray, grayish yellow, weak, broken, and gravel-like with a fragmented core; its joint fissures are filled with mud and are very developed, irregular, and open; it has a rough surface and a large amount of iron and manganese. | The third-stage slope | It is protected by polymer curing agent, system anchor, water-absorbing polymer materials, and external-soil spray-seeding technique. |
| Hunan test slope | Intense weathered granite | It has a sand–mud structure with micro-expansion property; it has a large amount of mica and vertically developed joint fissures; it has serious differential weathering (spherical weathering, sac-like weathering, etc.); its liquid limit index is too large; it has low strength, poor stability, and low damage-resistance. | The first, second and third stage slopes | It is protected by polymer curing agent, system anchor, water-absorbing polymer materials, and external-soil spray-seeding technique. |

## 4. Results and Discussion

### 4.1. Performance of Polymer Curing Agent

#### 4.1.1. Workability Performance

Test results of viscosity are shown in Figure 7.

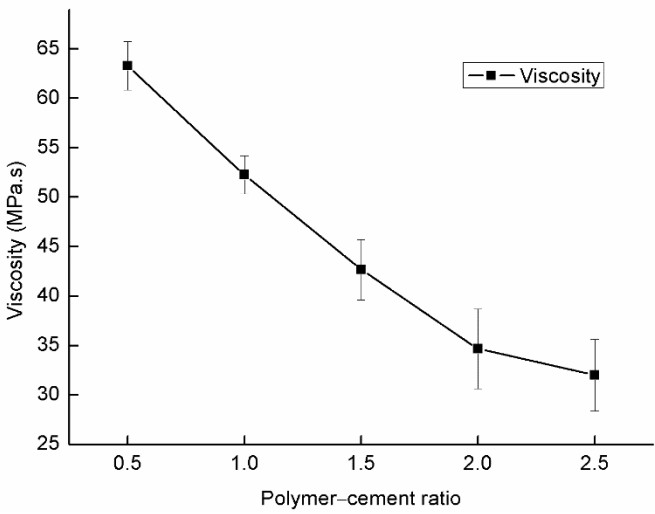

**Figure 7.** Relationship between polymer–cement ratio and viscosity.

As shown in Figure 7, it can be calculated that the viscosity of the polymer curing agent decreases by 49% as the polymer–cement ratio increases. When the polymer–cement ratio is greater than 2, the change of viscosity tends to be stable as the polymer–cement ratio increases. With the increase of the polymer–cement ratio, the content of the polymer emulsion increases and cement particles decrease relatively. When the polymer–cement ratio reaches a certain limit, the change of viscosity tends to be stable. It was found that the above mix proportions had good fluidity and met the requirements of construction spraying in the viscosity test.

#### 4.1.2. Bonding Performance

Test results of bonding strength at 7 d and 28 d are shown in Figure 8.

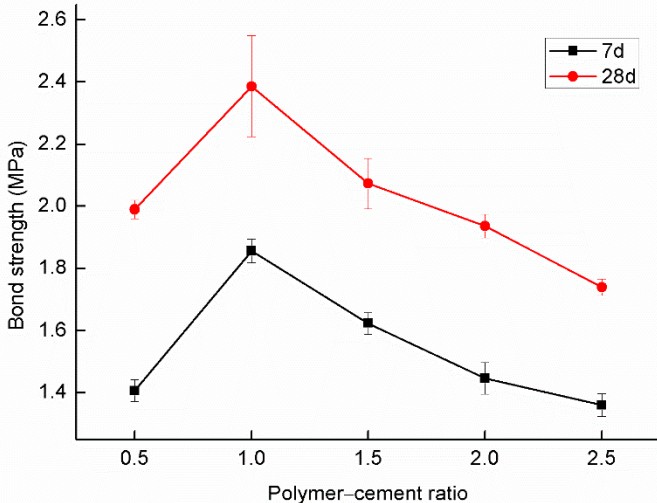

**Figure 8.** Test results of bonding strength of different polymer–cement ratios.

Through the analysis of test results in Figure 8, the bonding strength of the polymer curing agent does not increase as the polymer–cement ratio increases. As the polymer–cement ratio increases, the bonding strength of polymer curing agent increases first and then decreases. When the polymer–cement ratio is 1, the bonding strength of the polymer curing agent reaches the maximum. Because the chemical reaction between the polymer organic groups in the polymer emulsion, solid calcium hydroxide, and silicate on the surface of the cement hydration products is the most complete, the bonding ability of the cement hydration products is improved. When the polymer–cement ratio is greater than 1, it continues to increase so that the relative content of cement decreases. At this time, the bonding strength is mainly generated by the polymer emulsion. When the polymer–cement ratio is smaller than 1, the bonding strength of the layer begins to decrease. The main reason is that with the decrease of the polymer–cement ratio and the increase of the cement, the cement gradually replaces the latex and its hydration products play a leading role. The bonding strength at 28 d is much larger than that at 7 d because the hydration and polymerization of the cement in the emulsion is more complete with the extension of the reaction time. The bonding strength is mainly generated by the gelation of cement and polymer polymerization, latex in the emulsion, and hydration products of cement, etc.

### 4.1.3. Waterproof Performance

Test results of water-absorbing rate are shown in Figure 9.

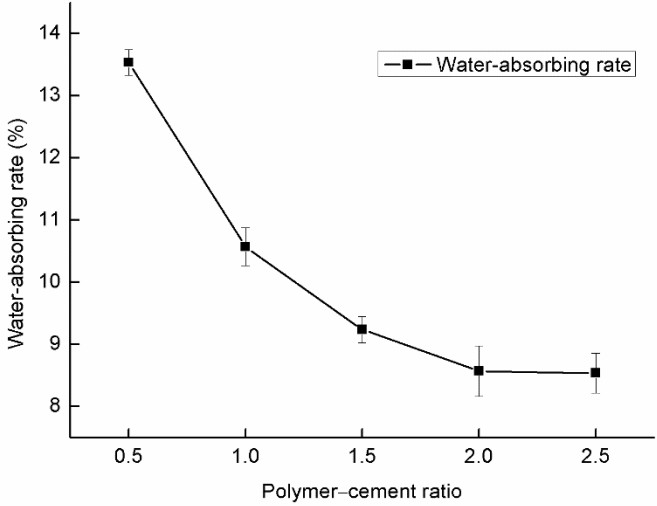

**Figure 9.** Relationship between polymer–cement ratio and water-absorbing rate.

In Figure 9, the water-absorbing rate of specimen is relatively large when polymer–cement ratio is relatively small. However, it gradually decreases and tends to be stable as the polymer–cement ratio increases. The reason is that when the polymer–cement ratio is relatively small, less polymer emulsion and more cement results in more hydration products with network structure. A small amount of polymer emulsion cannot completely fill the network structure, so the relatively small layer is generated on the polymer network structure. Thus, the water-absorbing rate of the specimen is relatively large. As the polymer–cement ratio continues to increase, the polymer emulsion is dominant in the polymer curing agent, causing the polymer material to react with the hydroxyl groups on the surface of the weak rock with a waterproof layer formed. Therefore, the water-absorbing rate gradually decreases until it tends to be stable.

### 4.1.4. Hydrophobic Performance

Measurement results of the surface contact angle of the specimen before and after curing are shown in Figure 10.

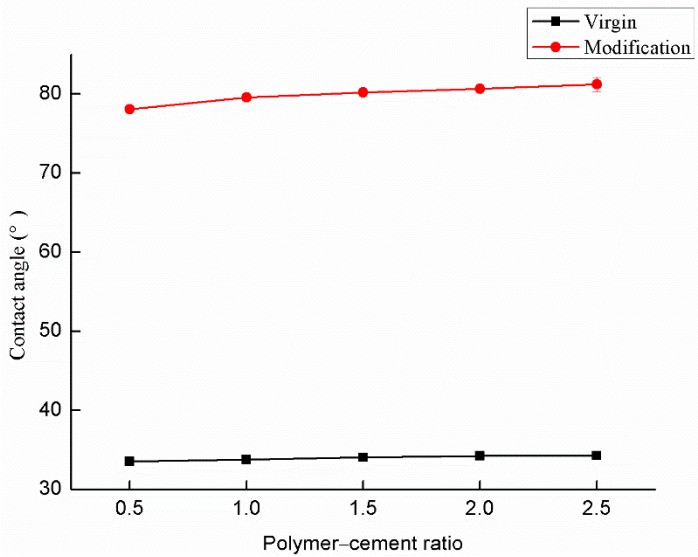

**Figure 10.** Comparison of contact angle before and after modification.

It is seen in Figure 10 that, as the polymer–cement ratio increases, the contact angle gradually increases, which then tends to become stable. In addition, the polymer emulsion increases and the cement decreases. The hydrophobic layer of polymer latex is mainly formed on the surface of polymer curing agent so that the wetting angle increases. When the polymer–cement ratio further increases, the hydrophobic performance of the polymer curing agent is dominantly affected by the polymer emulsion so that it tends to be the maximum. Through the comparative analysis of the wetting angle before and after modification, the polymer curing agent can improve the hydrophobic performance of the weak rock surface. The main reason is that a part of water in the polymer emulsion is volatilized, so that polymer particles are dehydrated and bond together to form a continuous elastoplastic layer. On the other hand, after the cement absorbs the remaining water in the emulsion, it is hydrated and polymerized. After the reactions, it is cured and forms a waterproof film structure of an interpenetrating network together with the organic polymer chain. Some gaps exist in the molecules of the solid polymer. The width of the gap is about several tens of nanometers. It is reasonable that a single water molecule can pass through these gaps, but water by nature is usually in an association condition. Due to the action of hydrogen bonds, a larger cluster of water molecules is formed among the polymer molecules.

### 4.1.5. Microscopic Performance

With the curing agent of different polymer–cement ratios, the photomicrographs of the weak rock surface before and after curing are shown in Figure 11.

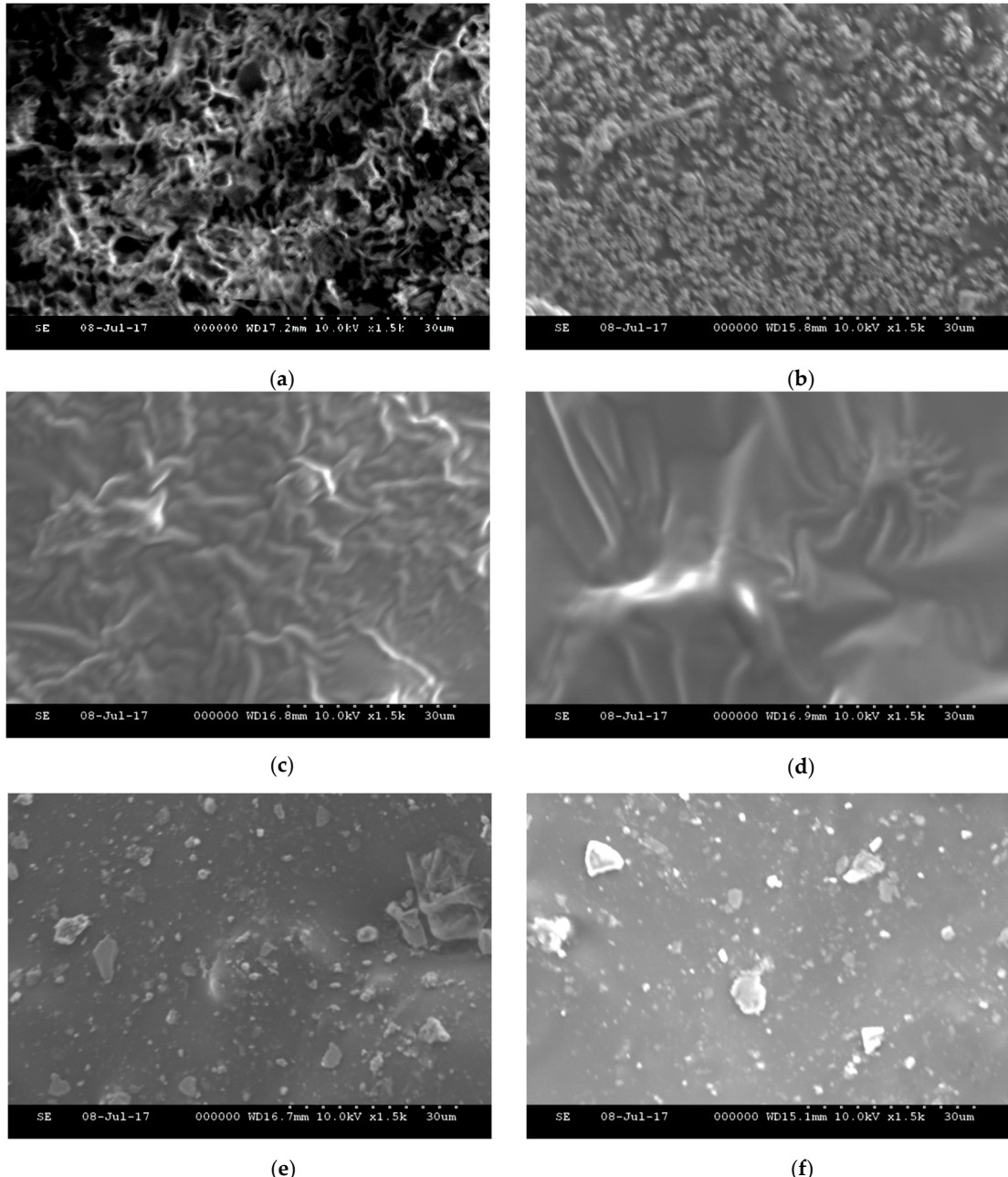

**Figure 11.** Micrographs of the weak rock before and after curing at different polymer–cement ratios: (**a**) P/C = 0; (**b**) P/C = 0.5; (**c**) P/C = 1.0; (**d**) P/C = 1.5; (**e**) P/C = 2.0; (**f**) P/C = 2.5.

Based on SEM micrographs, different morphologies are generated on the weak rock surface as the polymer–cement ratio increases. It can be observed that when the polymer–cement ratio is 1, the polymer phase and the cement phase are interpenetrated, cross-linked, and cured. The formed interpenetrating network structure has both a flexible network of organic polymer materials and a network structure of inorganic cementing materials. In addition, when the polymer–cement ratio is less than 1, pores of the weak rock surface increase. It is mainly because more hydration products are generated and their particles gradually cluster in the capillary pores. A close-packed layer is then

formed on the surface of the gel and on the incompletely hydrated cement particles and these clustering particles of hydration products gradually fill these pores. When the polymer–cement ratio is greater than 1, the cement particles are completely encapsulated by the polymer emulsion. A part of water in the polymer emulsion is volatilized so that the polymer particles are dehydrated and bonded together, thereby forming a continuous elastoplastic layer, which is mainly composed of a polymer emulsion. It is indicated that the polymer–cement ratio affects the microstructure of the weak rock surface, which is an important factor affecting the waterproofness and hydrophobicity of the polymer curing agent.

*4.2. Test Section*

Combined with engineering economy, the polymer curing agent has better comprehensive protection performance when the polymer–cement ratio is 1 in this test. Therefore, the polymer curing agent with the polymer–cement ratio of 1 is applied to the weak rock test slope of the highways in Zhejiang and Hunan. After the construction of the test slope, the following conditions were observed.

4.2.1. Construction Observation

As shown in Figure 12, the preparation work of construction of the polymer curing agent for the weak rock slope was directly sprayed by using the existing external-soil spray-seeding equipment.

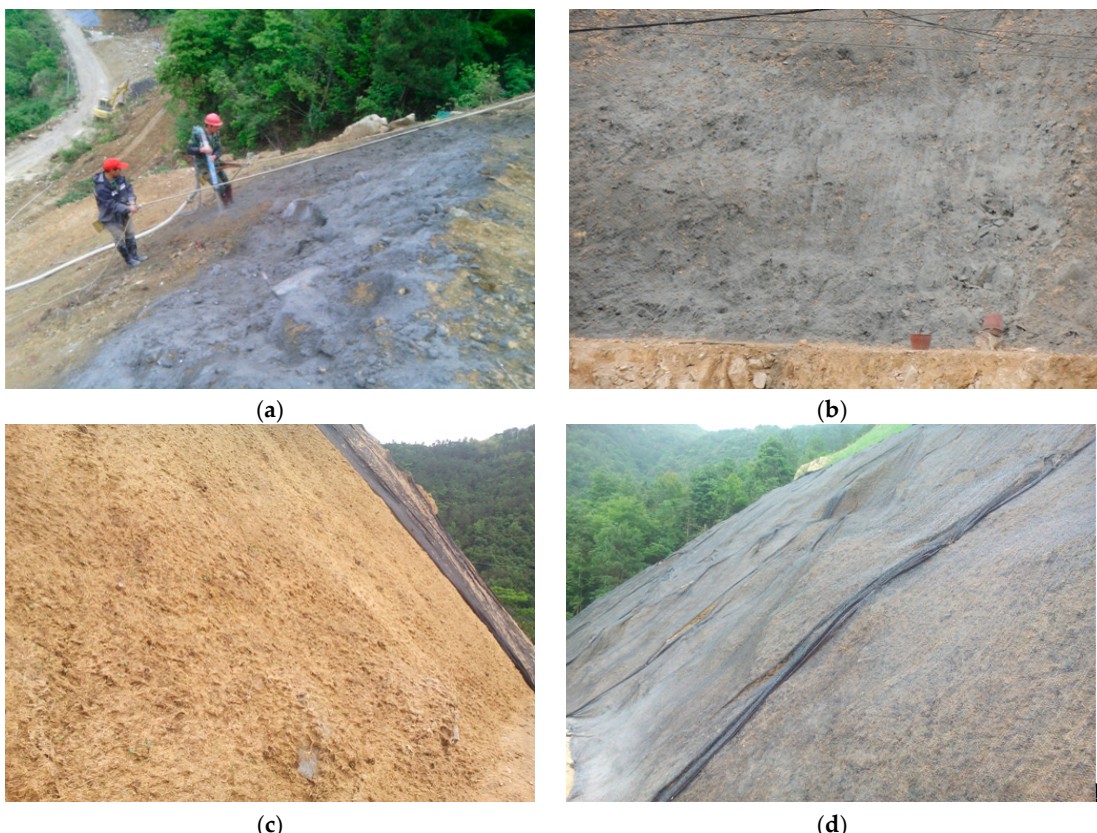

**Figure 12.** Construction of polymer curing agent for weak rock slope: (**a**) Spraying; (**b**) construction of wire mesh; (**c**) external-soil spray-seeding; (**d**) curing.

4.2.2. Observation of Curing Effect

An effective protective layer was formed on the slope surface by using the polymer curing agent, which achieved a good bonding and integration effect. As shown in Figure 13, after spraying, the effect of polymer curing agent layer was observed.

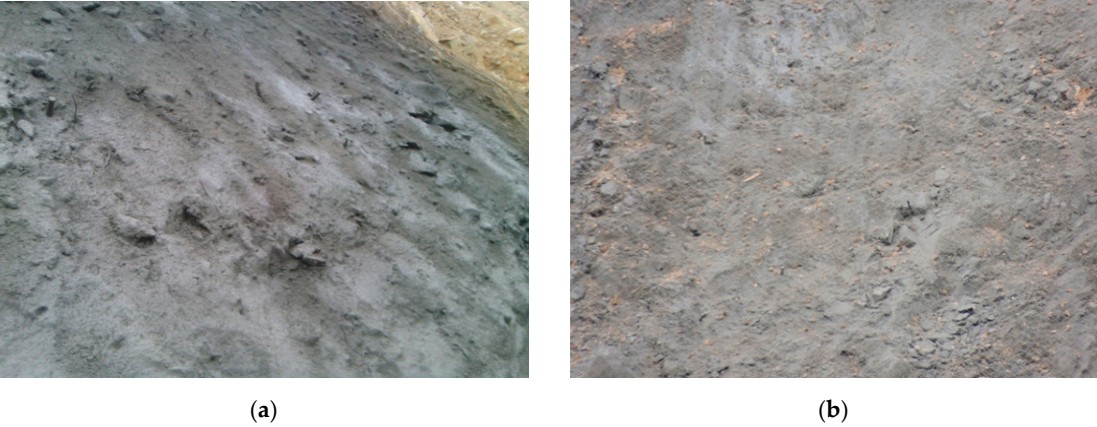

| (**a**) | (**b**) |

**Figure 13.** Curing effect: (**a**) Closed void; (**b**) loose particles of bonded surface.

### 4.2.3. Observation of Growth Condition of Plant Root System

The growth of the flora root system was not prevented or hindered by the polymer curing agent layer. The plant root system could penetrate the curing agent layer and enter the slope for ecological protection, as shown in Figure 14.

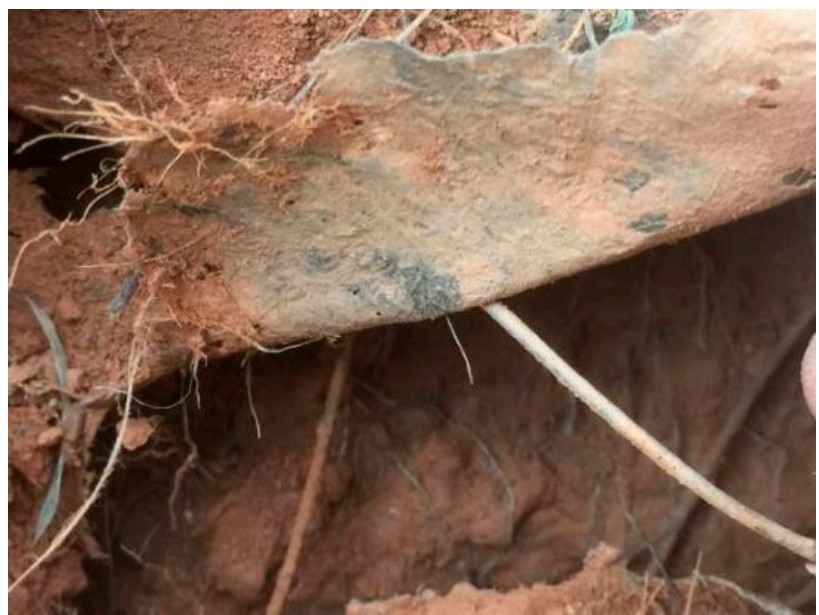

**Figure 14.** Penetration of the plant root system into the curing agent layer.

### 4.2.4. Observation of Greening Effect of Test Slope

From the greening effect diagrams of Figures 15 and 16, it was found that most shrubs and grasses grow well and the plants on the test slope treated with the curing agent were more flourishing than that of the test slope without the curing agent. The reason is that rainwater and fertilizer water on the slope with the curing agent can penetrate into the curing agent through the external soil layer. Due to the water impermeability of the curing agent layer, the water preserving capability of the external soil, and the water absorption of polymer, the rainwater and the fertilizer water were stored in the polymer water-absorbing materials of the external soil and the pores of the surface of the curing agent layer, which was required for the growth of plants. Water and nutrients could be directly absorbed when plants need them. Therefore, the plants grew faster. However, the weak rock slope without the curing agent layer could be directly infiltrated by the rainwater and the fertilizer water through the external

soil layer. Water and nutrients were still in a dispersed state when plants needed them. Therefore, plants grew slowly.

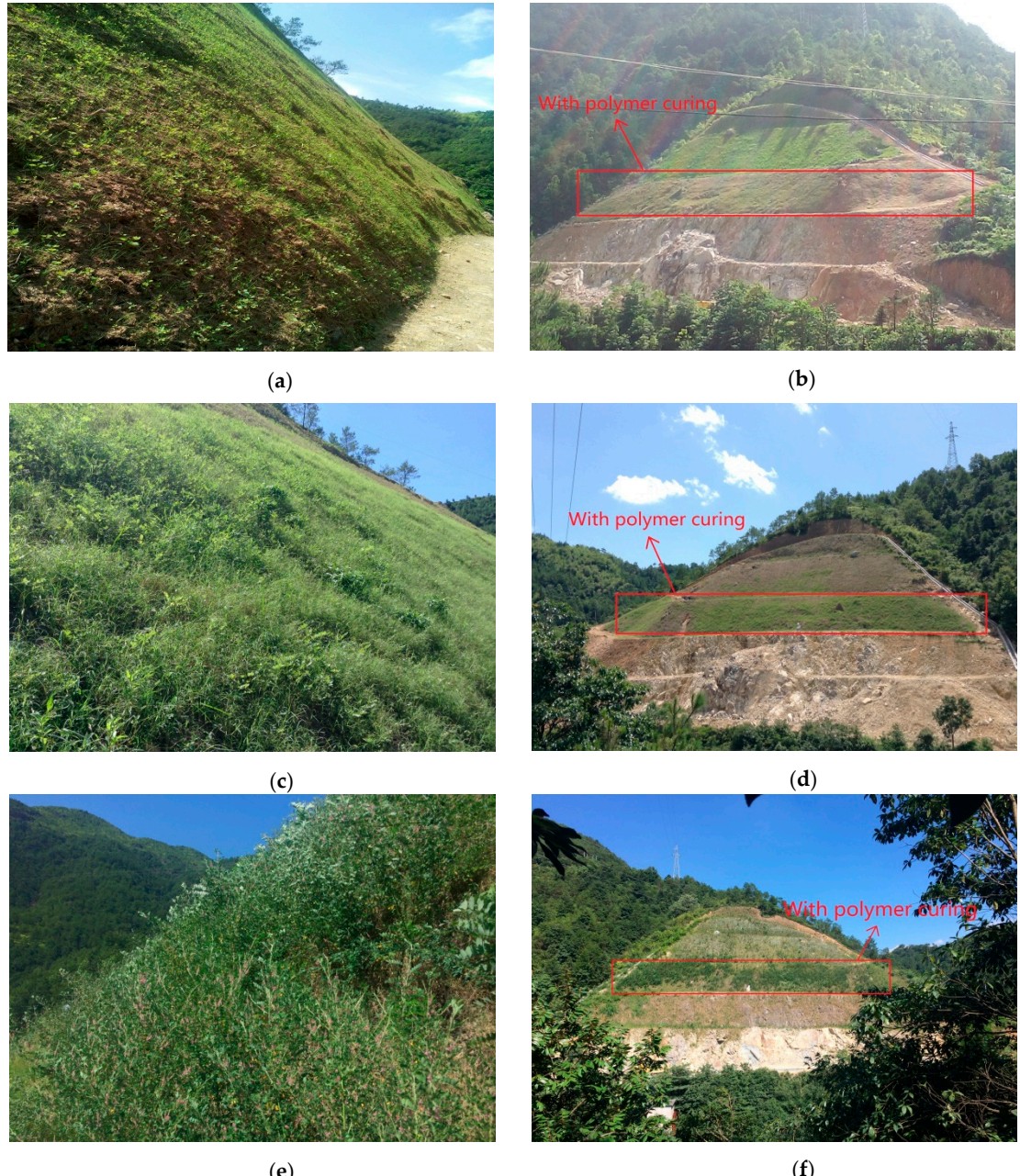

(**a**)　　　　　　　　　　　　　　　　　　　(**b**)

(**c**)　　　　　　　　　　　　　　　　　　　(**d**)

(**e**)　　　　　　　　　　　　　　　　　　　(**f**)

**Figure 15.** Plant growth conditions of the Zhejiang test slope: (**a**–**b**) 1 month later; (**c**–**d**) 3 months later; (**e**–**f**) 1 year and 3 months later.

The ecological protection of the weak rock slope is a new kind of protection technology. Combined with the projects in Zhejiang and Hunan, the relevant practical techniques summarized by the project team are as follows.

(1) Construction process

(a) In the rainy season, construction was carried out using the following process: Cleaning the slope → curing of the weak rock slope (spraying the curing agent) → construction of the slope anchor → hanging and fixing the wire mesh → spraying the curing agent again (complementing spraying, preventing omission, and spraying the curing agent layer damaged in construction) → external-soil spray-seeding (adding controllable polymer water-absorbing materials) → curing.

(b) In the dry season, construction was carried out using the following process: Cleaning the slope → construction of the slope anchor → curing of the weak rock slope (spraying the curing agent) → hanging and fixing the wire mesh → spraying the curing agent again (complementing spraying, preventing omission, and spraying the curing agent layer damaged in construction) → external-soil spray-seeding (adding controllable polymer water-absorbing materials) → curing.

(2) The traditional external-soil spraying machine was used as the spraying equipment of the polymer curing agent.

(3) When the polymer–cement ratio of polymer curing agent was 1, an effective protective layer was formed on the slope surface and good adhesion and integration effects were achieved.

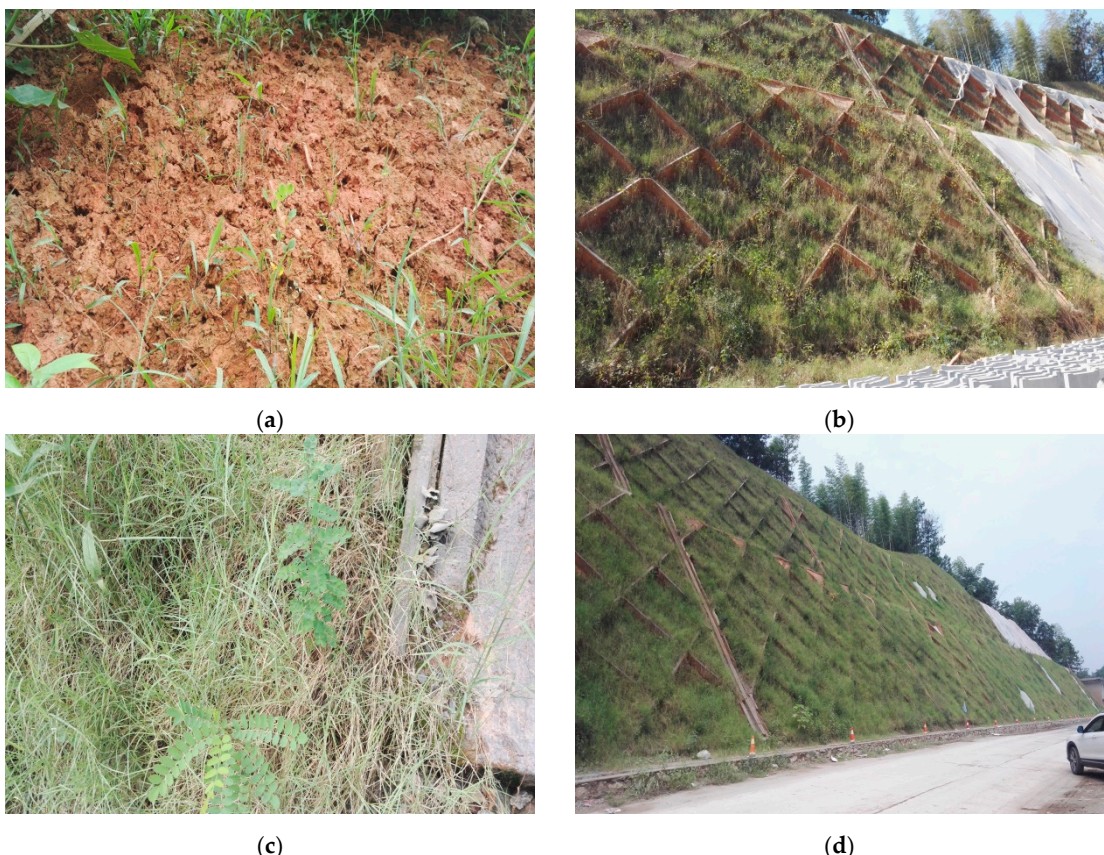

(**a**)                                    (**b**)

(**c**)                                    (**d**)

**Figure 16.** Plant growth condition of the Hunan test slope: (**a**) 1 month later; (**b**) 3 months later; (**c–d**) 6 months later.

## 5. Conclusions

Through the performance analysis, it has been found that the ecological protection of the polymer curing agent has many advantages, which provides a reference for similar projects. However, slopes with different geological conditions have different ecological protection methods. Therefore, when the ecological protection method in this study is applied in similar slope protection projects, local geological conditions should be taken into consideration.

(1) The stability safety factor of the weak slope under the protection of the polymer curing agent was analyzed by the strength reduction method. It was found that after the weak rock slope was treated by the polymer curing agent in this project, the safety factor of the slope stability increased.

(2) On the basis of laboratory and on-site experiments, the effects of the polymer curing agent on the weak rock slope were studied. From the laboratory test results, it can be seen that the strength, water stability, and surface microstructure of the weak rock slope were improved by the polymer curing agent layer. A curing layer with certain hydration resistance and softening resistance on the

surface of excavated weak rock slope can be formed. From the on-site test results, it is shown that the weak slope surface has high hydration resistance and softening resistance. It can meet the requirements of the curing treatment of the weak rock slope.

(3) From laboratory tests, it is observed that the change of properties of the curing agent depends on the polymer–cement ratio and curing time. The bonding strength increases as the curing time increases. From SEM images, it is indicated that the polymer curing agent has an interaction between the polymer emulsion, cement, and soft rock. These interactions have greatly changed the surface structure and physical and chemical properties of soft rock, thereby improving the strength, water stability, and erosion resistance of the slope surface.

(4) From on-site tests, it is indicated that the polymer curing agent of the weak slope is beneficial to improve the water-damage resistance of the slope surface and prevent or reduce the softening of weak rock, thus allowing the greening of the weak rock slope. Through this research, it is clearly pointed out that the styrene–acrylic emulsion cement-matrix composite material could be used as a new type of polymer curing agent for weak slope ecological protection engineering. This technique is worth applying in weak slope ecological protection.

**Author Contributions:** Data curation, D.Y. and J.L.; investigation, D.Y. and J.Y.; methodology, D.Y., J.L. and J.Y.; writing—original draft, D.Y. and J.L.; writing—review and editing, G.Q. and J.Y.

**Funding:** This study was supported by the National Natural Science Foundation of China (51778071), Transportation Science and Technology Project of Zhejiang Province, China (grant number 2015J17) and Transportation Science and Technology Project of Hunan Province, China (grant number 201505).

**Acknowledgments:** We express our sincere gratitude to Zhengxiang Zhou of Department of Economic Management, Changsha University of Technology, and anonymous reviewers for their constructive comments.

**Conflicts of Interest:** We declare no conflict of interest.

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
