# Peer review of "Application of Polymer Curing Agent in Ecological Protection Engineering of Weak Rock Slopes"

_applsci, doi:10.3390/app9081585_

Round 1
Reviewer 1 Report
The Mohr-Coulomb relationship should be expressed in terms of effective normal stress on the shearing plane. In the FEM analysis should be specified the angle of dilatancy.
Moreover, the paper would certainly benefit from a proofread since there are minor grammatical errors and a few spelling mistakes.
Author Response
Dear Reviewer:
Thank you for the reviewer’ comments concerning our manuscript entitled “Application of Polymer Curing Agent in Ecological Protection Engineering of Weak Rock Slope”. (ID: applsci-475614). Those comments are all valuable and very helpful for revising and improving our paper, as well as the important guiding significance to our researches. We have studied comments carefully and have made correction which we hope meet with approval. Special thanks to you for your good comments.
Response to Reviewer 1 Comments
Point 1: The Mohr-Coulomb relationship should be expressed in terms of effective normal stress on the shearing plane. In the FEM analysis should be specified the angle of dilatancy.
Response 1: We have made correction according to the Reviewer’s comments. Based on the associated flow rule,the dilatancy angle is Ψ =φ in the FEM analysis. The value of the dilatancy angle under the condition of non-associated flow rule is the next step of our research.
Point 2: The paper would certainly benefit from a proofread since there are minor grammatical errors and a few spelling mistakes.
Response 2: We have proofread the grammatical errors and some spelling errors in the full text.

Reviewer 2 Report
As the manuscript deals with cement-based material and discussed polymer curing agent, it is highly recommend to read and cite following papers:
a- Chemical Engineering Journal, 345: 471-482 (2018)
b- Construction and Building Materials, 190: 1264-1283 (2018)
2- Please present the conclusion without the numbers and bullet points. In its language layer, the submitted manuscript must be double checked.
Author Response
Dear Reviewer:
Thank you for the reviewer’ comments concerning our manuscript entitled “Application of Polymer Curing Agent in Ecological Protection Engineering of Weak Rock Slope”. (ID: applsci-475614). Those comments are all valuable and very helpful for revising and improving our paper, as well as the important guiding significance to our researches. We have studied comments carefully and have made correction which we hope meet with approval. Special thanks to you for your good comments.
Response to Reviewer 2 Comments
Point 1: As the manuscript deals with cement-based material and discussed polymer curing agent, it is highly recommend to read and cite following papers:
a- Chemical Engineering Journal, 345: 471-482 (2018)
b- Construction and Building Materials, 190: 1264-1283 (2018)
Response 1: According to the reviewer's suggestion, we have added the literature.
Point 2: Please present the conclusion without the numbers and bullet points. In its language layer, the submitted manuscript must be double checked.
Response 2: We have made correction according to the Reviewer’s comments. We have proofread the grammatical errors and some spelling errors in the full text.

This manuscript is a resubmission of an earlier submission. The following is a list of the peer review reports and author responses from that submission.
Round 1
Reviewer 1 Report
In reviewer's opinion the topic of the paper is interesting, but several issues must be clarified before considering the paper for a possible publication in the journal.
a) The proposed solution should be discussed in more details highlighting advantages and limitations of the methodology respect to other ones existing in literature.
b) 2.4.6. Test Section: this section needs to be expanded and explained in more details with a better organizing of the case studies and a better presentation of the results obtained. Moreover, the authors could offer a broader view of their solution in the context of the available literature on the topic. This would help highlighting the relevance of the work.
c) Paragraph 3.2.4: This paragraph needs to be expanded with a better organizing and presentation of the main influencing factors of mountain stability in the study area.
d) Abstract, introduction and conclusions should be better presented.
Minor corrections
1) In the caption of Figure 1 insert a blank space between (a) and virgin.
2) In the caption of Figure 7 insert a blank space between Figure 7. and Construction.
3) For completeness the following references could be added (page 3, line 72) after the statement “Water is one of main factors of instability of weak rock slope”:
Iverson R. M. (2000). Landslide triggering by rain infiltration. Water Resources Research, 36(7), 1897-1910.
Dai F. C., Lee C. F., Wang S. J. (2003). Characterization of rainfall-induced landslides. International Journal of Remote Sensing, 24 (23), 4817-4834.
Conte E., Donato A., Pugliese L., Troncone A. (2018). Analysis of the Maierato landslide (Calabria, Southern Italy). Landslides, 15(10), 1935-1950. DOI: 10.1007/s10346-018-0997-x
Reviewer 2 Report
The submitted manuscript deals with the use a styrenic-acrylic emulsion as protective agent in weak rock slope. The application seems interesting, but the approach is definitively poor from a scientific point of view, so it does not deserve to be published. An emulsion is not a curing agent!
All the experimental results (bonding strength in Figure 3, water absorbing rate in Figure 4, contact angle values in Figure 5 need to be absolutely revised by introducing error bars). On the other hand, SEM images of the different surfaces in Figure 6 have no sense at all (what the authors have seen in Figure 6d??? In addition, (e)P/C=2; and (f)P/C=2.0 are related to the same or a different material?